# Splay-bend nematic phases of bent colloidal silica rods induced by polydispersity

Ramakrishna Kotni[1,2], Albert Grau-Carbonell [1,2], Massimiliano Chiappini[1,2], Marjolein Dijkstra [1] ✉ & Alfons van Blaaderen [1] ✉

Liquid crystal (LC) phases are in between solids and liquids with properties of both. Nematic LCs composed of rod-like molecules or particles exhibit long-range orientational order, yielding characteristic birefringence, but they lack positional order, allowing them to flow like a liquid. This combination of properties as well as their sensitivity to external fields make nematic LCs fundamental for optical applications *e.g.* liquid crystal displays (LCDs). When rod-like particles become bent, spontaneous bend deformations arise in the LC, leading to geometric frustration which can be resolved by complementary twist or splay deformations forming intriguing twist-bend ($N_{TB}$) and splay-bend ($N_{SB}$) nematic phases. Here, we show experimentally that the elusive $N_{SB}$ phases can be stabilized in systems of polydisperse micron-sized bent silica rods. Our results open avenues for the realization of $N_{TB}$ and $N_{SB}$ phases of colloidal and molecular LCs.

In 1949 Onsager predicted that, at sufficiently high densities, fluids of hard-rod-like particles align along a common nematic director $\hat{\mathbf{n}}$[1], thereby yielding an entropy-driven phase transition from an isotropic (*I*) to a nematic (*N*) phase characterized by long-range orientational order but bereft of any positional order. Although being derived for infinitely long rods, Onsager's predictions found confirmation in computer simulations on rods of large but finite length[2,3]. Abundant experimental evidence has since then followed[4–8]. The entropy-driven nature of the isotropic-to-liquid crystal phase transitions of hard particles makes it especially susceptible to subtle changes in the particle shape and symmetry that are hard to predict based on intuition. For instance, over 3 decades ago, it was shown by computer simulations that hard rods with a spherocylindrical shape do form smectic phases, which are 1-dimensional (1D) long-range ordered liquid crystal phases composed of parallel layers of orientationally aligned rods forming 2D liquids, while hard rods with an ellipsoidal shape do not[9]. A more recent example can be found for systems of hard brick-like particles, for which it has been theoretically predicted to promote a so-called biaxial nematic ($N_B$) phase in which both particle axes are simultaneously aligned along two orthogonal nematic directors pointing in the direction of long-range orientational order[10]. However, the stabilization of this biaxial nematic phase turned out to be hindered by a

competition with a positionally one-dimensionally (1D) ordered smectic (*Sm*) phase[11–16]. Biaxial nematic phases can indeed be stabilized in colloidal LCs by destabilizing the smectic phase using polydispersity[17,18]. The experimental observation of stable biaxial nematic phases of molecular LCs has been historically challenging. In 2004, $N_B$ phases were reported for systems of so-called bent-core mesogens, which are, because of the similarity in shape, also referred to as banana-shaped LCs[19–21]. However, such observations remain disputed to this date. In contrast, biaxial nematic phases have been reported for goethite brick-shaped particles[22], in accordance with theoretical and computational predictions[16,23,24]. Furthermore, biaxial smectic-A phases[25], intimately related to biaxial nematic phases and also holding potential for applications such as optical switching have been reported as well for molecular[26] and colloidal systems[27].

It was once believed that banana- or boomerang-shaped molecules would not be compatible with the formation of liquid crystal phases. However, studies that started almost a century ago showed this not to be the case[28] and the LC phase behavior of molecules or particles with bent-core shapes is now known to be truly rich[29,30], although these first studies hardly had any impact for over 60 years. It was realized about 50 years later that the frustration brought about by rods with a bend could bring about interesting new nematic liquid

---

[1]Soft Condensed Matter, Debye Institute for Nanomaterials Science, Utrecht University, Princetonplein 1, 3584 CC Utrecht, The Netherlands. [2]These authors contributed equally: Ramakrishna Kotni, Albert Grau-Carbonell, Massimiliano Chiappini. ✉e-mail: M.Dijkstra@uu.nl; A.vanBlaaderen@uu.nl

crystal phases[31,32] in which the fact that spontaneous bend deformations in the nematic director field cannot extend over long distances is reconciled with complementary splay or twist deformations, yielding $N_{SB}$ or $N_{TB}$ nematic phases, respectively[31,33]. An $N_{TB}$ nematic phase is characterized by spontaneous breaking of chiral symmetry (even for achiral molecules) and an $N_{SB}$ nematic phase by alternating splay and bend domains, making these "frustrated" nematic phases of interest to arrive at fundamental studies, but also for applications as the different length scales associated with these phases allow for many new possible applications[34,35]. Schematics of the $N_{SB}$ and $N_{TB}$ phases, in which the average particle orientations vary in space yielding a continuum nematic director field $\hat{\mathbf{n}}(\mathbf{r})$ with head to tail invariance, are depicted in Fig. 1a. Despite numerous experimental realizations of stable $N_{TB}$ phases in systems of bent-core mesogens[36,37], an experimental observation of an $N_{SB}$ phase was still lacking more than 40 years after its first theoretical prediction in 1976[31]. After finishing this work we became aware of a recent paper in which $N_{SB}$ phases of molecular LCs are

reported, but they were induced by electric fields[38] in agreement with theoretical predictions[39]. Very recently, however, computer simulations showed that the stability of the $N_{SB}$ phase is hindered by an overly stable positionally ordered Sm phase, and that smooth curvature in the particle shape and polydispersity in the length distribution of bent particles could be two experimentally realizable approaches to lift the "smectic blanket" and reveal stable $N_{SB}$ phases[40]. Interestingly, a colloidal model system from an organic material with a smooth particle curvature was recently developed, and quasi-2D samples of these smoothly curved particles display indeed an $N_{SB}$ phase[41]. In this paper, we consider bent silica rods that strongly favor smectic phases because of the interlocking of the sharp particle kinks. Here, we follow the approach of destabilizing the smectic phase by introducing polydispersity to stabilize the $N_{SB}$ phase. We employ polydisperse bent silica rods, which is based on a relatively recently developed new colloidal silica rod-shaped model system[42,43] that can be imaged quantitatively on the single-particle level. Already it has been shown that by

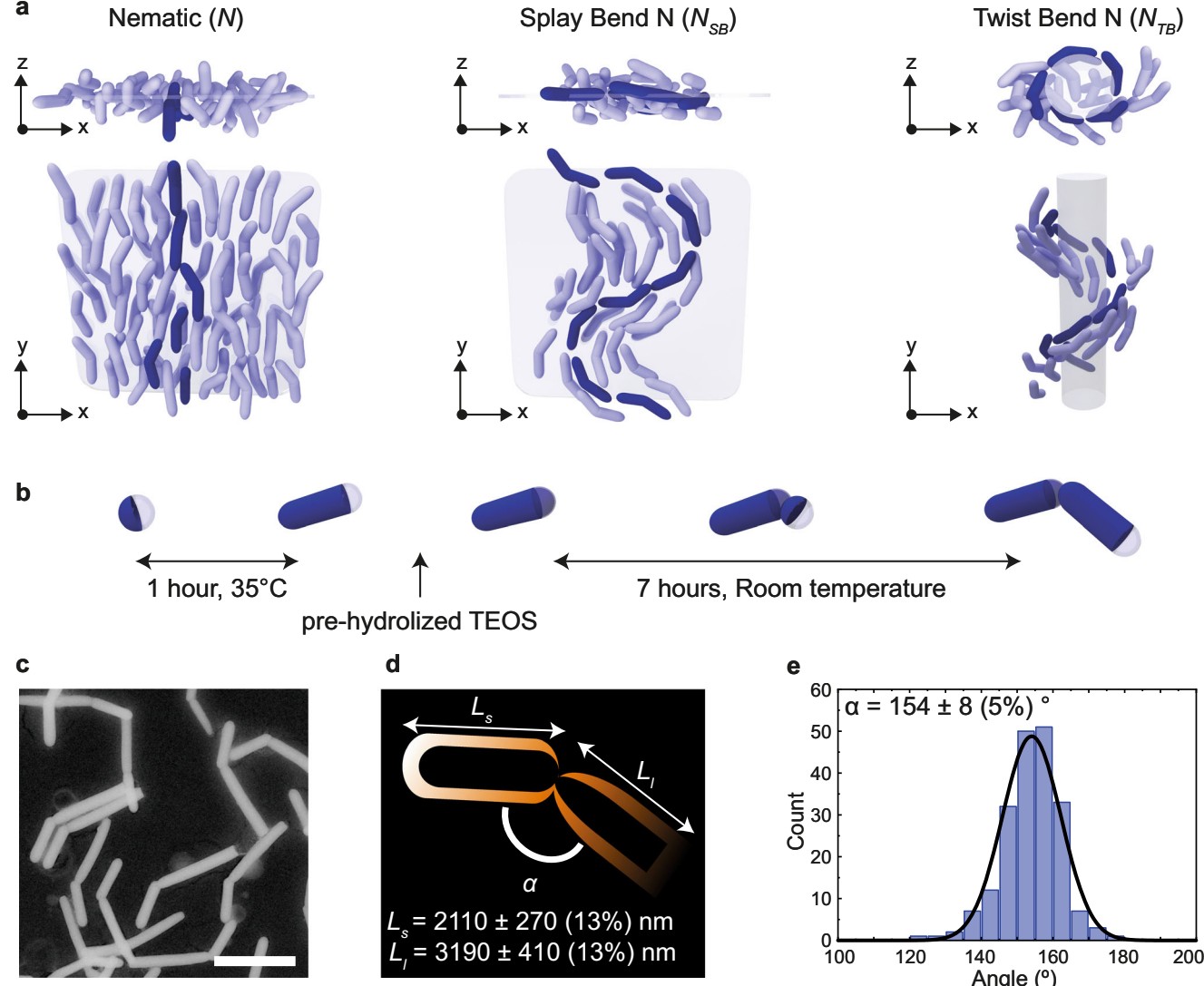

**Fig. 1 | Schematics of LC phases and characterization of a system of bent silica rods (BSRs). a** Side and top schematic views of a uniaxial *N*, splay-bend $N_{SB}$, and twist-bend $N_{TB}$ nematic liquid crystal phase of bent-core rod-like particles. The darker particles highlight the change in orientation of the nematic director along the sample, which extends to a 3D modulation in the $N_{SB}$ and $N_{TB}$ phases. The *xy* and *xz* planes are indicated as will be defined further for confocal microscopy.

**b** Synthesis process of the BSRs, consisting of a first growth step at 35 °C followed by an injection of pre-hydrolized TEOS and 7 more hours of growth to finish the reaction. **c** Scanning transmission electron microscopy of BSRs (scale bar 3 μm). **d** Schematic of the density distribution of the fluorophores along the BSRs and characterization of the length of the short ($L_s$) and long ($L_l$) segments of the BSRs. **e** Distribution of bend angles ($\alpha = 154 \pm 8°$) of the BSRs.

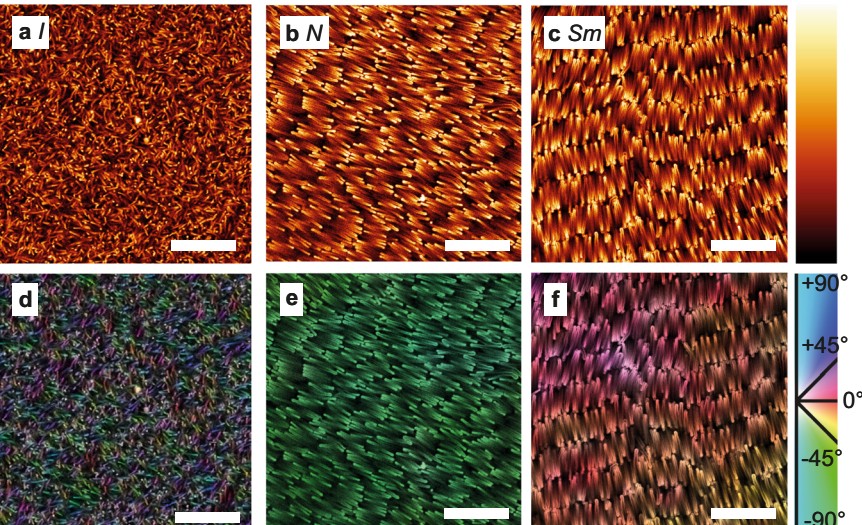

**Fig. 2 | Liquid crystal phases of fluorescently labeled bent silica rods (BSRs) from a sedimentation-diffusion equilibrium profile, as imaged with fluorescence confocal microscopy. a** Isotropic (*I*) phase (above $14 \pm 1\,\mu m$ from the bottom surface), **b** Nematic (*N*) phase (between $8 \pm 1\,\mu m$ and $14 \pm 1\,\mu m$) and **c** Smectic (*Sm*) phase (from the bottom surface to the *N* phase). On the right, color code for image generation, brighter parts indicate higher signal and thus fluorophore density. **d**–**f** Orientational analysis of the *I*, *N*, and *Sm* phases, respectively. The color code indicates the local orientation as obtained from an analysis of the local pixels. Scale bars: 10 μm.

fluorescently labeling these silica rods it becomes possible to image not only their position, but also the orientation of these anisotropic rods even in nematic and smectic LC phases[7,44,45]. Moreover, when these rods are sterically stabilized with short-linear alkane ligands these systems can be refractive-index matched and the interactions between the charged and sterically stabilized systems can be tuned from almost hard-rod-like to long-range repulsive to an extent where the silica rods form a plastic crystal, which is the "opposite" phase from a liquid crystal in the sense that for these solids positional order is long-range but the rotational degrees of freedom are disordered[46,47]. Both the plastic crystals and LC phases could be manipulated and ordered over large distances using electric fields[46,48], which is interesting for applications similarly as it is for their molecular counterparts. Additionally, even more recently an other group[49,50] and our group[51] were able to modify the synthesis of the rods so that also bent-core silica particles can be made. All these recent advances allowed us to realize a colloidal model system for which a splay-bend $N_{SB}$ phase was realized and that can now be studied on a single-particle level. The possibility to fully characterize a $N_{SB}$ system on a single-particle level in 3D that can be manipulated by external fields opens the door to directly study the effects of size, shape, angles and their corresponding polydispersity on phase formation, switching and relaxation processes.

## Results and discussion

In this study, it was found that when the previous approaches to arrive at bent silica rods (BSRs)[49,51] were combined (Fig. 1b) a system of smooth, rigid core bent-shaped fluorescent silica colloids could be synthesized (Fig. 1c). See SI for more details on the procedure used. As a result of the synthesis method the BSRs incorporated more dye molecules at the start of the reaction, resulting in a non-homogeneous distribution of fluorescence levels (Fig. 1d). We were able to realize and observe an $N_{SB}$ phase by studying a sedimentation-diffusion profile of fluorescent BSRs with sufficient intrinsic polydispersity in diameter ($D = 490 \pm 100$ nm (20%)), length ($L_s = 2110 \pm 270$ nm (13%) and $L_l = 3190 \pm 410$ nm (13%)), and angle ($\alpha = 154 \pm 8°$ (5%), see Fig. 1e). Interestingly, the polydispersity in length of our system (13%) was significantly lower than that described for a $N_{SB}$ phase by computational methods (36%)[40]. However, our experimental system consists of

building blocks with polydispersity also in the angle between particle segments, the length of each segment and their width. Therefore, it is likely that for experimental systems the smectic phase could be destabilized by contributions from all different polydispersities combined. Confocal microscopy was used to characterize the self-assembly of fluorescent BSRs as a function of height for differences in the LC phases formed along the sedimentation-diffusion profile. We characterized the $N_{SB}$ phase by identifying its unique structure at the single-particle level and obtaining its characteristic "wavy" nematic director field. We utilized computationally generated fluorescence confocal-like data stacks of $N_{SB}$ and $N_{TB}$ phases to confirm the qualitative and quantitative accuracy of our analysis methodology and results. Furthermore, we extended the synthesis of our bent silica rods to design bent rods with single-crystalline corundum hematite ellipsoidally shaped seeds, which also showed a $N_{SB}$ phase thereby opening the door to the assembly of $N_{SB}$ phases with functional colloidal building blocks and additionally manipulate their self-assembly both by external electric and magnetic fields.

Suspensions of BSRs were left to sediment and equilibrate a sedimentation-diffusion profile for several days in DMSO/water (78% DMSO) index matched ($n = 1.45$, $T = 20\,°C$) solvent in which silica colloids are stable due to their surface charge. Salt (0.64 mM LiCl) was used to set a well-defined electric double-layer thickness ($\kappa^{-1} = 10$ nm)[52]. Then, the samples were mounted on a confocal microscope and imaging of the phase behavior was performed as a function of height. When a sedimentation-diffusion equilibrium is reached in a colloidal dispersion of hard rods, a number of phases establish as a function of height in the sample, corresponding to the local osmotic pressure and density[53]. Here, at the uppermost parts of a sedimentation-diffusion profile of BSRs, around $14 \pm 1\,\mu m$ above the bottom surface, an isotropic (*I*) phase was identified (Fig. 2a). A nematic (*N*) phase was found between $8 \pm 1\,\mu m$ and $14 \pm 1\,\mu m$ above the bottom surface (Fig. 2b), whereas an *Sm* phase was found between the bottom surface and the *N* phase (Fig. 2c). Confocal microscopy images were analyzed for their local structure at the single-particle level by investigating the structure tensor of every pixel. This local orientation was visualized as a color map, as seen for the *I* phase (Fig. 2d), *N* phase (Fig. 2e) and *Sm* phase (Fig. 2f). Particle orientations were homogeneous in the nematic regions, whereas the smectic regions showed a

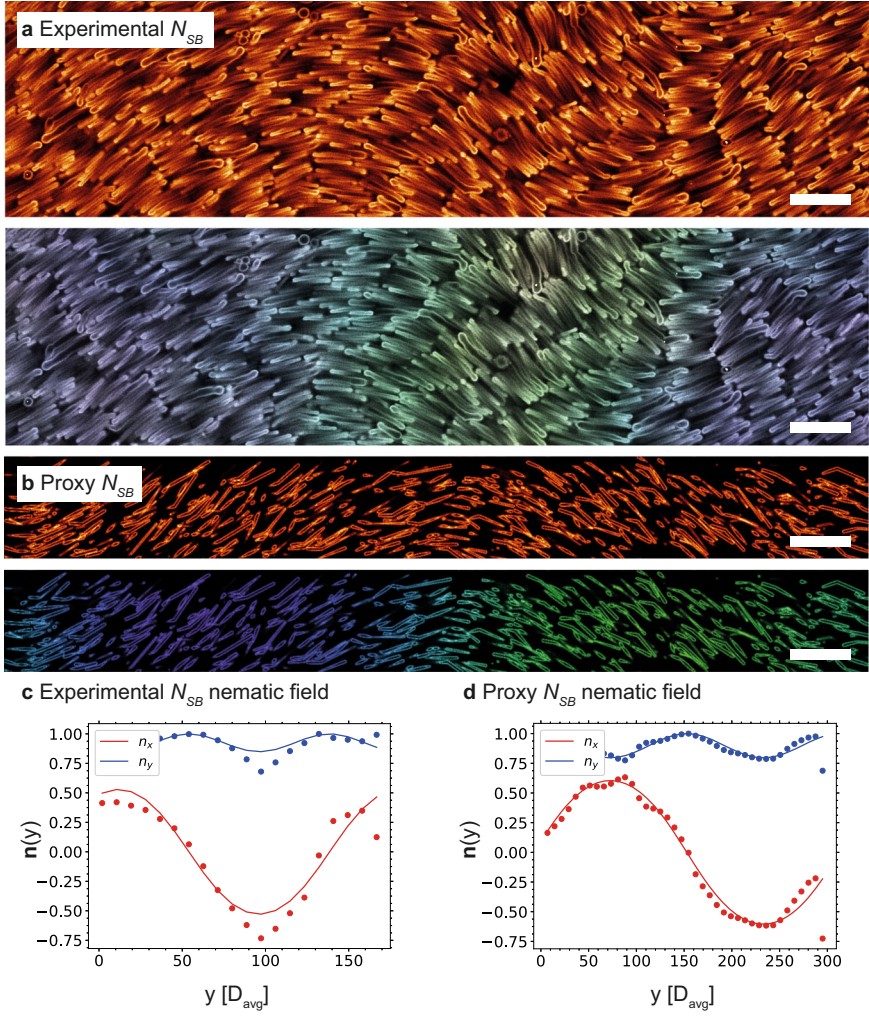

**Fig. 3 | Characterization of the $N_{SB}$ phase. a** Fluorescence confocal microscopy image of a $N_{SB}$ domain located between the previously identified $N$ and $Sm$ phases (between $4 \pm 1\,\mu m$ and $8 \pm 1\,\mu m$ from the bottom surface). **b** Proxy, computationally generated fluorescence confocal microscopy image of a dispersion of rods in a $N_{SB}$ configuration. **c** Characterization of the nematic director field of the experimental $N_{SB}$ phase. The extracted values were $p = 86.2 \pm 11.7\,\mu m$ and $\theta_0 = 0.46 \pm 0.01$ rad. Spatial coordinates are given in terms of the average diameter of the particles

($D_{avg}$). Images were binned to make $D_{avg}$ as close as possible to 6 pixels to match the experimental data to the computationally generated data. **d** Characterization of the nematic field of the proxy $N_{SB}$ phase. The extracted values were $p = 89.9 \pm 1.1\,\mu m$ and $\theta_0 = 0.64 \pm 0.01$ rad, which are representative of the parameters used to generate the phase: $p = 84.2\,\mu m$ and $\theta_0 = 0.59$ rad. Spatial coordinates are given in terms of the average diameter of the particles ($D_{avg}$). Source data are provided as a Source Data file. Scale bars: 5 μm.

degree of swirliness. It is possible that some of the smectic regions are resulting from a frustrated $N_{SB}$ phase as mentioned in a paper on computer simulations[40], and retained part of the modulations of the $N_{SB}$ phase, perhaps until further equilibration into an unperturbed smectic.

In polydisperse systems of crooked rod-like particles, $N_{SB}$ phases are expected to be found in a pressure range in between that of a $N$ and $Sm$ phase[40]. In our experimental system, we indeed found an $N–N_{SB}$ transition above the $Sm$ phase, between $4 \pm 1\,\mu m$ and $8 \pm 1\,\mu m$ above the bottom surface, with the $N_{SB}$ displaying periodic modulations in the long-range orientational order of the bent silica rods (Fig. 3a). Such $N_{SB}$ domains were found across multiple samples (Supplementary Figs. 1–4). As our $N_{SB}$ phase was formed in a gravitational field, the modulations of the long-range order of the nematic director field was found to be in planes perpendicular to that of gravity and thus in planes parallel to the $xy$ imaging plane of our confocal set up. We were thus interested if the modulation identification and discrimination between an $N_{SB}$ and related $N_{TB}$ phase could be accomplished by analyzing 2D planes only. To investigate this, proxy ad hoc $N_{SB}$ (Fig. 3b) and $N_{TB}$ (Supplementary Fig. 5) phases were computationally

generated and confocal 3D stacks were simulated using a theoretical point spread function for a confocal microscope as a means to confirm our phase identification and analysis from 2D confocal images only. The details about the generation and analysis of these proxy phases can be found in the Supplementary Information. It can be readily seen that the $N_{SB}$ and the $N_{TB}$ phases were fundamentally different under simple visual inspection as well, the latter presenting periodic regions with particles oriented perpendicularly to the imaging plane independently of the orientation of the phase. The 2D cuts of the experimental $N_{SB}$ phase reported here match the structure of the proxy $N_{SB}$ phase. Additionally, we checked that the modulation pattern as abstracted from the generated images were close to those as used to generate the structures (SI). Thus, we analyzed the modulation pattern of our experimental confocal images in the same way.

To confirm the identification of the elusive $N_{SB}$ phase beyond any reasonable doubt, we performed an orientational analysis of Fig. 3a, resulting in a vector field, which maps the local orientation of the rods in the images and reflects the 2D nematic director field of the phase. The resulting nematic director field of the experimental phase of Fig. 3a was well fit by the theoretical nematic director field of a $N_{SB}$

phase $\hat{\mathbf{n}}(\mathbf{y}) = (\sin(\theta_0 \cdot \sin(qy)), \cos(\theta_0 \cdot \sin(qy)), 0)$, the $y$-axis corresponding to the global nematic director, i.e., the average particle orientation of the whole phase, and $\theta_0$ being the maximum angle of oscillation of the particle orientation, $q = 2\pi/p$ the wave number of these periodic oscillations, and $p$ the pitch length. This good description confirmed the experimental determination of the $N_{SB}$ phase (Fig. 3c). This analysis was repeated for all $N_{SB}$ phases (shown in Supplementary Figs. 1–4), and yielded an average measure of the amplitude and periodicity of the splay-bend modulations $\theta_0 = 0.55 \pm 0.11$ rad and $p = 86.2 \pm 11.7\,\mu m$. Interestingly, the pitch of the $N_{SB}$ phase was several times larger than the total (end-to-end) particle size, showcasing the presence of multiple length scales in the self-assembled system and thereby making it a promising system for optical applications as well. To confirm that the pitch and amplitude of the phase could be reliably characterized in this way, the same analysis was also performed on the proxy $N_{SB}$ phase (Fig. 3d). We found that the fitting parameters extracted from the analysis of the proxy $N_{SB}$ phase $\theta_0 = 0.64 \pm 0.01$ rad and $p = 89.9 \pm 1.1\,\mu m$ were consistent with the known values $\theta_0 = 0.59$ rad and $p = 84.2\,\mu m$, thereby confirming that $N_{SB}$ phases could indeed be reliably characterized with our methodology.

As already discussed, the orientational order of nematic phases can be characterized by continuum director fields with head to tail invariance $\hat{\mathbf{n}}(\mathbf{r})$[54]. At places where $\hat{\mathbf{n}}$ is not uniquely defined, like in a point or line in an $N$ phase, a nematic defect is present. At finite temperatures entropy dictates that defects are always present in condensed phases of matter in equilibrium, but they are often also out-of-equilibrium manifestations of the formation of the phase that did not have enough time to anneal out. For instance for LCs they may stem for instance from symmetry breakings in different directions and different regions remaining from an $I$–$N$ transition[55]. It is not surprising therefore that we also observed many defect structures for which we now have the opportunity to study these on the single-particle level, which is not trivial for ordinary LCs[56]. Line defects can be categorized by their disclination strength ($S$), which is defined as the number of times the nematic director rotates by $2\pi$ for a close path in the plane perpendicular to the line defect. Examples of line defects with disclination strength $S = \pm 1/2$ are presented in Fig. 4[56]. Such defects were commonly found in multiple samples (Supplementary Fig. 6).

After the successful creation of an $N_{SB}$ phase from BSRs we realized that even more interesting $N_{SB}$ phases from both a fundamental and application point of view might be realized if the BSR synthesis could be combined with a seeded growth procedure that was already developed for the kind of silica rods we used[57]. As proof of principle we combined the BSRs with single-crystalline hematite ellipsoids, which in a magnetic field have a magnetic dipole moment in the direction of the long axis of the ellipse[58] and combined these seeds with non-fluorescent BSRs (BSRs, Fig. 5a). As mentioned, this particle system was synthesized via a seeded growth method[57], followed by the previously described BSRs synthesis procedure, and consisted of particles with ($D = 281 \pm 33$ nm (12%), $L_s = 1315 \pm 84$ nm (6%), $L_l = 2570 \pm 390$ nm (15%)) and $\alpha = 151 \pm 9°$ (6%)). This hematite-BSR system is special in several ways: The hematite seed particles used were single-crystalline corundum seeds with an ellipsoidal shape. The seeded growth of the bent silica rods was initiated by droplet attachment to only one side of the ellipsoid, which means that the magnetic dipole is induced in a direction perpendicular to the long axis of the ellipsoid, is directed parallel with the bent rod director. This opens up the possibility to modify the self-assembly of these colloidal BSRs not only with electric fields but independently also with magnetic fields[59,60], which is of interest both for fundamental studies, but also for photonic applications as the hematite ellipsoids are strongly scattering particles. At the bottom of the sedimentation-diffusion profile, we were able to identify a $Sm$ via confocal microscopy, by collecting the reflected laser signal at 495 nm (signal mainly from silica) and 670 nm (signal mainly from the hematite ellipsoids) (Fig. 5b). However, the particles were not fluorescently labeled and imaging at higher positions in the sediment was therefore not possible. We thus allowed our samples to slowly dry and analyzed the resulting pellets via scanning electron microscopy (SEM). Figure 5c, d show SEM images of the dried pellets. It can be observed that the internal structures seen are compatible with an $N_{SB}$ phase, as revealed by the wavy modulation pattern, along which some of the structure also cracked because of drying induced stresses. The pitch of the modulation was again 10–15 times larger than the particle size. Preliminary experiments showed that dispersions of BSRs could be manipulated with external electric fields (Supplementary Fig. 7).

Using polydispersity to realize $N_{SB}$ and/or $N_{TB}$ phases is a general strategy that can be applied to other systems as well. A preliminary promising example is shown in Fig. 5e where a wavy pattern pointing towards a tendency to form a $N_{SB}$ phase can be recognized in this dried self-assembled thin phase of boomerang-shaped brookite TiO$_2$ nanorods ($D = 7.1 \pm 0.8$ nm (12.6%), $L = 82.9 \pm 10.4$ nm (12.6%)) simply by drying the nanorods on a TEM grid[61]. We finally, expect and suggest that $N_{SB}$ phases could be realized for LCs of bent-core mesogens by combining molecules with different lengths to obtain polydisperse systems. Mixtures of bent-core molecular liquid crystals and mixtures of straight and bent-core LCs were already studied to extend and/or shift transition temperatures[62], but as far as we know

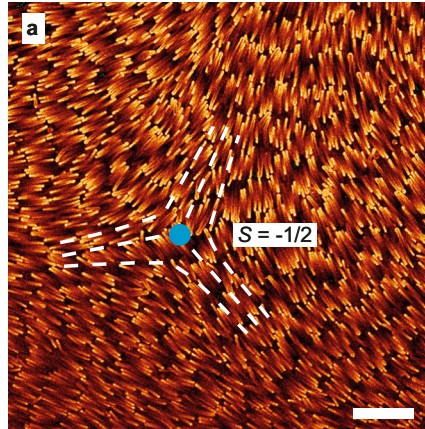
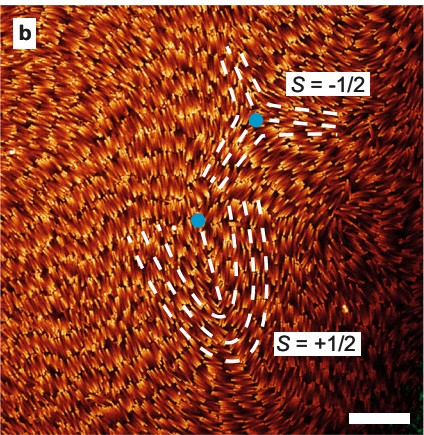

**Fig. 4 | Fluorescence confocal microscopy images of defects in nematic phases of BSR.** Blue dots indicate the position of the defect and dashed white lines give an indication of the direction of the nematic director field. **a** Line defect with disclination strength $S = -1/2$. **b** Pair of line defects meeting each other with disclination strengths $S = \pm 1/2$. Scale bars: 10 μm.

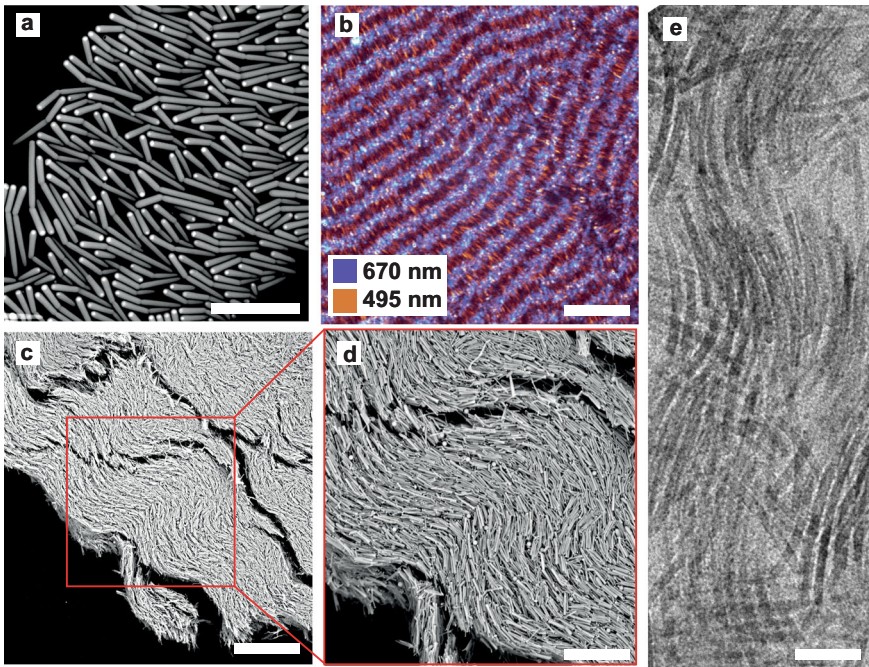

**Fig. 5 | Phase behavior of hematite core-shell BSRs (CS-BSRs). a** (S)TEM image of CS-BSRs. The hematite ellipsoid was located at one end of the rod with its long axis perpendicular to the rod length as the CS-BSR were obtained via a seeded growth methodology. **b** Smectic domain as imaged by reflection confocal microscopy. **c** Dry pellet of CS-BSRs displaying an internal structure compatible with an $N_{SB}$ phase. **d** Close-up image showcasing the modulation in the nematic director along the pellet. **e** Brookite titania bent nanorods dried onto a TEM grid showcasing a tendency towards the formation of an $N_{SB}$ phase. Scale bars: **a** 2 µm, **b** 10 µm, **c** 30 µm, **d** 20 µm, and **e** 20 nm.

not with the intention to realize $N_{SB}$ phases. Thus, we believe our approach is easily applicable to systems over many length scales. Additionally, it has recently been shown that LC phases with regions with more bent and splay are interesting to spatially separate differently shaped colloids[63], which would also be interesting to investigate on the single-particle level.

To conclude, we reported on the experimental realization and identification of an $N_{SB}$ phase from polydisperse bent-core fluorescently labeled colloidal building blocks that could be analyzed on a single-particle level. The bent silica rods were made using a modification of an existing general emulsion-based growth method that is also compatible with seeded growth. The $N_{SB}$ was characterized via the analysis of its nematic director field from an analysis of fluorescence confocal micoscopy data sets, and yielded a pitch of $p = 86.2 \pm 11.7$ µm, i.e., about 16 times the particle length. The $N_{SB}$ phase was found to form in between a *Sm* and an uniaxial *N* phase, as expected from theory and simulations[29,30,40,64,65]. $N_{SB}$ LC phases were also observed for a system of hematite functionalized bent silica rods oriented perpendicular to the single-crystalline hematite ellipsoids that can be magnetized in magnetic fields with a dipole moment in the direction of the ellipsoids. Our results together with those on polymer-based smoother bent rods developed independently from us[41], suggest that using polydispersity is indeed a viable and generally applicable methodology to create these frustrated $N_{SB}$ liquid crystal phases. Additionally, the feasibility of creating such interesting "frustrated" nematic liquid crystal phases also from more complex shaped bent silica rods like the bent silica rods with well oriented hematite cores in addition to preliminary results on bent titania nanorods make it likely in our opinion that $N_{SB}$ phases might also be formed from mixtures of bent molecular mesogens even without external electric fields[66,67]. Further work has to demonstrate if also twist-bend nematic phases can be realized in these systems. As this will allow for single-particle-based studies, it may resolve for instance recent controversies around these experimentally more abundant phases[66,67].

## Methods

### Materials

Polyvinylpyrrolidone (PVP, Mn = 40 kg/mol), 1-pentanol (>99%, Sigma-Aldrich), ammonia solution (NH$_3$, 25 wt% in water, Sigma Aldrich), anhydrous ethanol (>100%, Interchema), sodium citrate tribasic dihydrate (>99%, Sigma Aldrich), tetraethylorthosilicate (TEOS, >99%, Sigma-Aldrich), (3-aminopropyl)-triethoxysilane (APTES, MERCK, >98%), 7-nitrobenzo-2-oxa-1,3-diazol (NBD, 98%, Across organics), FeCl$_3$.6H$_2$O (97%, Sigma-Aldrich), sodium hydroxide (NaOH) (pellets, Merck), hydrogen chloride (HCl, 37 wt.%, ACS reagent), dimethylsulfoxide (DMSO, >99.8%, Sigma Aldrich).

### Synthesis of bent silica rods (BSRs)

In a typical synthesis, 1.0 g of PVP (25 µmol) was dissolved in 10.0 mL 1-pentanol (92 mmol) using sonication (Branson Bransonic CPXH Digital Bath 8800, Emerson) for 1 h. After the complete dissolution of PVP in 1-pentanol, a clear solution was formed. To this solution, ethanol (1.0 mL, 17.4 mmol), distilled water (0.280 mL, 15 mmol), and an aqueous solution of sodium citrate (67.0 µL, 0.18 M) were added in sequence. The reactants were homogenized by shaking the flask with hand for 3–4 times. Then, 0.225 mL of ammonium hydroxide (1.7 mmol ammonia, 9.4 mmol water) was added and the reactant mixture was mixed by continuous shaking for 1 min to create an emulsion. To this emulsion, TEOS (0.10 mL, 0.45 mmol) was added and the reaction mixture was kept at 35 °C for 1 h, Subsequently, 15.0 µL (0.06 mmol) of partially hydrolyzed TEOS was added. All the ingredients were homogenized by carefully tilting the bottle side to side for 3–4 times and the vial was transferred to room temperature and left to react for 7 h. Eventually, the obtained rods were centrifuged at 1200 × *g* for 10 min and dispersed in absolute ethanol. This washing step was repeated two times. The partially hydrolyzed TEOS was prepared as follows: 5.0 mL TEOS (22.3 mmol) was mixed with 11.0 µL HCl (0.44 mmol) and 0.4 mL distilled water (22.2 mmol) and vortexed (MS2 Minishaker, IKA) for 1 min.

## Particle labeling with NBD for fluorescence confocal microscopy imaging

For fluorescent labeling, we first reacted a silane coupling agent (APTES) covalently with the dye (NBD) in ethanol, which we refer to as NBD-APTES solution. In a typical procedure, 18.0 μL (0.06 mmol) of NBD was dissolved in 5.0 mL absolute ethanol (85.6 mmol) in a 30 mL laboratory vial and sonicated for 5 min. To this NBD ethanol solution, 35.0 μL (0.15 mmol) of APTES was added and the solution was stirred slowly at room temperature overnight. The NBD-APTES solution was added to the above reactant mixture. In a typical synthesis of BSRs, 20.0 μL of NBD-APTES was added to the reactant mixture just before the addition of TEOS (0.10 mL, 0.45 mmol).

## Synthesis of hematite ellipsoids

Monodisperse hematite ellipsoids were synthesized by using the hydrothermal reduction method[58]. The typical synthesis procedure is as follows; In a 250.0 mL of a volumetric flask, (2.0 M) $FeCl_3.6H_2O$ solution was prepared by dissolving 54.0 grams of iron (III) chloride hexahydrate in 100 mL of water. To this, 100.0 mL (5.40 M) of NaOH solution was slowly added under continuous stirring till the solution reached to pH 7. As a result, a brownish solution was formed and this solution was stirred at high speed (158 g) for 30 min to ensure the homogeneous distribution of reactants. After stirring, a highly viscous gel was formed. Further, this gel was tightly sealed in an autoclave and immediately transferred to a preheated oven at 100 °C and kept for 3 days of aging, the solution was washed two times with water and finally disperse the obtained hematite ellipsoids in water.

## Synthesis of hematite core-shell BSRs (CS-BSRs)

To synthesize hematite CS-BSRs, 1.0 g PVP was dissolved in 10.0 mL of 1-pentanol using sonication or vortex for 1 h. To this solution, 6.0 mg of hematite ellipsoids (Long axis: 198 ± 24 (12.0%) nm, short axis: 143 ± 18 (12.5%) nm) was added, and the solution was sonicated for another 1 h. As a result, a clear red colored suspension was formed. Subsequently, the remaining reactants (i.e., 1.0 mL anhydrous ethanol, 0.280 mL water, and 67.0 μL of sodium citrate) were added in sequence. The reactants were mixed by carefully tilting the bottle side to side for 3–4 times to achieve an homogeneous distribution of the reactants. Then, 0.225 mL of ammonium hydroxide (1.7 mmol ammonia, 9.4 mmol water) was added and the solution was mixed by shaking the flask with hand for 1 min to form the emulsion. To start the growth of the silica rods, 0.10 mL of TEOS (0.45 mmol) was added. All the ingredients were homogenized by carefully tilting the bottle side to side for 3–4 times, and the reaction mixture was kept at 35 °C for the first 1 h. To this 10.0 μL partially hydrolyzed TEOS was added and allowed the reaction to continue for 7 h at room temperature. The particles were collected by three consecutive centrifugation (1200 × g for 10 min) and re-dispersion steps in ethanol. After final washing, particles were stored in ethanol.

## Sample preparation

BSR with a final concentration of 0.4 vol% were dispersed in a DMSO/water mixture (78% w/w) adjusted to a final refractive index $n = 1.45$ with a refractometer (Abbe refractometer, NAR-3T, Atago). 400 μL of such dispersion were placed in a home-made cell suitable for confocal microscopy. This cell was built from an end of a Pasteur pipette glued to a # 0 Glass-Menzel coverslip supported on a taylor made microscopy glass slide.

## Scanning confocal fluorescence microscopy

Confocal microscopy was performed with a Leica SP8 confocal microscope equipped with a white light laser and a 90x oil confocal immersion lens. The excitation wavelength was 495 nm, and the fluorescence signal was collected for the range between 500 and 650 nm. If the microscope was operated in reflection mode the wavelengths are indicated in the corresponding section.

## Electron microscopy

For TEM analysis, a Tecnai 20 electron microscopy operating at 200 kV, and for STEM analysis a Tecnai 20F electron microscope operating at 200 kV in scanning transmission electron microscopy (STEM) mode were used. Scanning electron microscopy measurements were performed by Phenom ProX.

## Proxy phases preparation

Proxy $N_{SB}$ and $N_{TB}$ with given amplitude and periodicity were realized by inserting in a simulation box non-overlapping crooked rods of shape randomly sampled from the BSR experimental shape distribution, each with a random position **r** and an orientation corresponding to the phase nematic director field $\hat{n}(\mathbf{r})$. The resulting configurations were then compressed via a rapid quench at high pressures using an *NPT* Monte Carlo simulation method. The nematic director field of the resulting proxy phase was then obtained from the particle orientations, and fitted with the theoretical expression of the nematic director field (see Fig. 3d, dashed lines) to measure $p$ and $\theta_0$.

3D confocal data sets were generated from the above-described $N_{SB}$ configurations. To obtain data sets representative of our experimental system we first generated ground truth stacks consisting of rods whose fluorescent signal was located at their surface. This was achieved by defining a backbone for the bent rods from the coordinates and particle orientations obtained from the *NPT* Monte Carlo compression step and then assigning values to the voxels found at a distance of a particle radius from such backbone. The rest of the voxels remained thus as voxels with no signal. The ground truth stacks were then convolved with a theoretical point spread function for a confocal microscope with the Huygens Professional software. Final simulated 3D confocal data sets were analyzed for structure exactly as experimental confocal data sets.

## Analysis orientation of the confocal microscopy images

Particle orientations as the color maps in Fig. 2d–f and Fig. 3a, b, and the 2D nematic director field as in Fig. 3c, d were obtained via a local gradient structure tensor analysis using the OrientationJ software, with Cubic Spline gradient and a local window of 30 pixels (~5 particle diameters).

## Data availability

The imaging data used in the figures of this manuscript have been deposited in the 4TU database [https://data.4tu.nl/account/home#/collections/6260499]. Source data are provided with this paper.

## Code availability

The code used to generate ground truth 3D data sets from particle coordinates has been deposited in the Github database [https://github.com/AlbertGrauCarbonell/BentRods].

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

## Acknowledgements

R. Kotni, and A. van Blaaderen acknowledge the Netherlands Center for Multiscale Catalytic Energy Conversion (MCEC), an NWO Gravitation program funded by the Ministry of Education, Culture and Science of the government of the Netherlands. A. Grau-Carbonell acknowledges funding from the European Research Council (ERC) via the ERC Consolidator Grant NANO-INSITU (grant no. 683076). M. Chiappini and M. Dijkstra acknowledge financial support from the EU H2020-MSCA-ITN- 2015 project "MULTIMAT" (Marie Sklodowska-Curie Innovative Training Networks) [project number: 676045]. M. Dijkstra also acknowledges funding from the European Research Council (Grant No. ERC-2019-ADG 884902 SoftML).

## Author contributions

R.K. performed synthesis and characterization of the colloidal systems. A.G. and R.K. performed self-assembly experiments. A.G. performed confocal microscopy. A.G. and M.C. generated simulated 3D confocal data sets. M.C. performed analysis of the data sets. A.v.B. and M.D. designed and supervised this research.

## Competing interests

The authors declare no competing interests.
