## [Peer Review File · Nature Communications]

REVIEWER COMMENTS

Reviewer #1 (Remarks to the Author):

The manuscript reports on an experimental discovery of a so-called splay-bend nematic phase in an artificial system of bent silica rods. The findings are interesting and the paper is suitable for Nature Comm. However, the authors are advised to correct the presentation of the past and current developments in the field of liquid crystals.

1. The so-called “first experimental observations of stable biaxial nematic phases... [19-21]” remain questionable, demonstrated by subsequent experiments, reviewed in Ref.23. In contrast to the twist-bend nematic, there is presently no clear evidence of the biaxial nematic in the bent-core materials.
2. Panov et al. 2010 paper, Ref. 26, could not be cited as an evidence that “bend deformations in the nematic director field cannot extend over long distances is reconciled with complementary splay and twist...” Panov et al. described unusual textures of a material that was later demonstrated by Cestari et al, Borshch et al, and Chen et al, to exhibit an NTB phase. It would be more appropriate to cite alongside Ref.25 the work of I. Dozov (Europhysics Letters 2001 Vol. 56 Issue 2 Pages 247-253), instead of Ref. 26 since Dozov developed the theory of the twist-bend and splay-bend nematics.
3. I am puzzled by the statement that Ref.29 presents “numerous experimental realizations of stable NTB phases”: I read Ref 29 again and again and find no evidence of the NTB phase in the reported materials. Here is the Abstract of Ref.29 which indicates the absence of any claims of NTB: “The compounds with an even-numbered spacer display, in addition to a nematic phase, various smectic phases whose nature depends on the ratio between the length of the terminal chain and the length of the spacer (n/m). A modulated SmA^* phase was found for the ratio $n/m = 1$ to 1.6 and a SmC /crystal G polymorphism emerges at $n/m = 1.7$. In contrast, all liquid-crystalline dimers with odd-numbered spacers display an intercalated $B6$ phase regardless of the chain length.”
4. The schemes of the $-1/2$ defects in Fig.4 are wrong: The director field does not diverge at the three tips; there are no letters “X” at the tips, please check any textbook on liquid crystals for a correct drawing.
5. Fig 5 caption: “aa” apparently means “an”

Reviewer #2 (Remarks to the Author):

First off, this manuscript read really well, and the topic is at the forefront of current research in condensed matter. The system interrogated by the authors, namely bent silica rods (BSR) that allow for dye labeling as well as the creation of responsive core-shell BSR structures, shows some intriguing properties that permit imaging at the single particle as well as the assembly level. Images confirm the formation of the elusive NSB phase by EM and FC imaging and tie well into theoretical considerations. The phase is expected between the N and the Sm phase and does so in the experimental system. The work is significant and should receive attention in the community. Therefore, this manuscript should be considered for publication in Nature Communication. There are, however, a few points the authors may want to consider:

(1) How critical are the overall dimensions of the BSR (i.e., aspect ratio and bent angle)? Do theory or experiment suggest any range for which the NSB phase would show/form?

(2) What is the range of polydispersity for which theory or experiment would show the NSB phase? In other words, how polydisperse must and can it be?

(3) Considering potential applications, it would be of tremendous value if the authors could make some more detailed remarks about how the community can translate these findings to molecular systems.

(4) The introductory section describes biaxial nematics but leaves the biaxial smectic-A phase formed by more blank- or brick-shaped molecules out.

Aside from these suggestions to provide a little more detail on some specific aspects of this work, I see no reason why this manuscript should not be published in Nature Communications.

Great work!

Reviewer #3 (Remarks to the Author):

This is a well-written paper describing the experimental observation of a splay-bend nematic phase in a system of polydisperse, bent silica rods. I am no experimentalist, so cannot sensibly judge the experimental procedures, but the background looks excellent to me and the analysis of the results

also looks careful and thorough, so the authors have certainly convinced me that they have found an example of this somewhat elusive phase. Furthermore the synthetic route looks very promising for future application.

As the authors point out, this is not a "first", in that a colloidal splay-bend nematic phase has already been observed (ref 33). The question is thus whether this observation is sufficiently novel to warrant publication in Nature Communications. I think that the power of the new technique is enough, in my view, to overcome this barrier and I would recommend publication after the authors have attended to the following points.

1) Could the authors say how they chose the dimensions, angles, etc. of their bent rods? My guess is that this sample happened to be the one that worked, but some clarity on this would be helpful. Even knowing the properties of samples that did not give this phase would be helpful.

2) It would be useful to know more about the properties of the neighbouring phases shown in fig. 2. For example, the smectic looks very swirly by eye! What sort of smectic is it? Indeed might it not be a smectic at all, but one of these liquid crystalline frustrated phases? More detail on this would be useful.

3) Just for the record, values of the order parameters would be helpful, certainly in the nematic and SB phases.

Reviewer #1:

The manuscript reports on an experimental discovery of a so-called splay-bend nematic phase in an artificial system of bent silica rods. The findings are interesting and the paper is suitable for Nature Comm. However, the authors are advised to correct the presentation of the past and current developments in the field of liquid crystals.

We thank the reviewer for the kind words on our work. The remarks of the reviewer have improved and corrected our overview of the developments in the field. Furthermore, Figure 4 is now much more accurate and better distinguishes the defect from the general directions of the nematic phases around the defect.

1. The so-called “first experimental observations of stable biaxial nematic phases... [19-21]” remain questionable, demonstrated by subsequent experiments, reviewed in Ref.23. In contrast to the twist-bend nematic, there is presently no clear evidence of the biaxial nematic in the bent-core materials.

We agree with the referee that in molecular liquid crystals the experimental observations are disputed, and as such they cannot be considered definitive proof of biaxial nematics from bent core materials. On the other hand, a biaxial nematic phase has been shown in a polydisperse colloidal system of brick-shaped goethite particles in E. van den Pol et al., *Phys. Rev. Lett.* **103**, 258301 (2009), and theoretically predicted in S. Belli, A. Patti, M. Dijkstra, and R. van Roij, *Physical Review Letters* **107**, 148303 (2011). Computational realizations of biaxial nematic phases of board-shaped particles were found in S. Dussi, N. Tasios, T. Drwenski, R. van Roij and M. Dijkstra, *Physical Review Letters* **120**, 177801 (2018) and S. D. Peroukidis and A. G. Vanakaras, *Soft Matter* **9**, 7419 (2013), and a biaxial nematic phases of polydisperse bananas was found in M. Chiappini, T. Drwenski, R. van Roij, and M. Dijkstra, *Physical Review Letters* **123** (6), 068001 (2019).

In order to avoid incorrectly suggesting that biaxial nematics of bent core materials have been realized, we have changed the text to reflect the current state of the literature better: biaxial nematics of brick-shaped particles have been described, while the reported observations of biaxial molecular nematics remain disputed. The following changes have been made to the first paragraph:

However, the first experimental observation of stable biaxial nematic phases of molecular LCs has been historically challenging and was not reported until 2004 for systems of so-called bent core mesogens which are, because of the similarity in shape, also referred to as banana-shaped LCs \cite{Madsen, Acharya, Dong}. \linebreak

Is now:

The experimental observation of stable biaxial nematic phases of molecular LCs has been historically challenging. In 2004 \textit{N} \textit{B} phases were reported for systems of so-called bent core mesogens which are, because of the similarity in shape, also referred to as banana-shaped LCs \cite{Madsen, Acharya, Dong}. However, such observations remain disputed to this date. In contrast, biaxial nematic phases have been reported for goethite brick-shaped particles \cite{vandenPol2009}, in accordance with theoretical and computational predictions \cite{Simone, Vanakaras2013, dussi2018hard}.

2. Panov et. al. 2010 paper, Ref. 26, could not be cited as an evidence that “bend deformations in the nematic director field cannot extend over long distances is reconciled with complementary splay and twist...” Panov et al. described unusual textures of a material that was later demonstrated by Cestari et al, Borshch et al, and Chen et al, to exhibit an N_{TB} phase. It would be more appropriate to cite alongside Ref.25 the work of I. Dozov (Europhysics Letters 2001 Vol. 56 Issue 2 Pages 247-253), instead of Ref. 26 since Dozov developed the theory of the twist-bend and splay-bend nematics.

In Ref. 26 by Panov et al. unusual periodic textures were described and were briefly discussed in terms of the splay, twist and bend elastic constants. However, we agree with the reviewer in that the structural discussion of how bend deformations extend over long distances is not discussed. The suggested paper (Dozov (Europhysics Letters, 56(2), 247-253 (2001)) explicitly discusses the solutions for bend deformations in terms of twist and splay and is indeed a substantially better reference for “bend deformations in the nematic director field cannot extend over long distances is reconciled with complementary splay and twist...”. Therefore, reference 26 is now replaced by the suggested reference:

Dozov, I. “On the spontaneous symmetry breaking in the mesophases of achiral banana-shaped molecules” Europhysics Letters, 56(2), 247-253 (2001).

3. I am puzzled by the statement that Ref.29 presents “numerous experimental realizations of stable NTB phases”: I read Ref 29 again and again and find no evidence of the NTB phase in the reported materials. Here is the Abstract of Ref.29 which indicates the absence of any claims of NTB: “The compounds with an even-numbered spacer display, in addition to a nematic phase, various smectic phases whose nature depends on the ratio between the length of the terminal chain and the length of the spacer (n/m). A modulated SmA^* phase was found for the ratio $n/m = 1$ to 1.6 and a SmC /crystal G polymorphism emerges at $n/m > 1.7$. In contrast, all liquid-crystalline dimers with odd-numbered spacers display an intercalated $B6$ phase regardless of the chain length.”

The authors thank the reviewer for this observation. Indeed ref. 29 describes the formation of $B6$ type phases from bent core mesogens and does not reference N_{TB} phases. As experimental realizations of N_{TB} are in fact reported, appropriate references are now provided for this statement:

Borshch, V. and Kim, Y.-K. and Xiang, J. and Gao, M. and Jakli, A. and Panov, V.P. and Vij, J.K. and Imrie, C.T. and Tamba, M.G. and Mehl, G.H. and Lavrentovich, O.D. “Nematic twist-bend phase with nanoscale modulation of molecular orientation” Nature Communications 4, 2635 (2013) DOI: <https://doi.org/10.1038/ncomms3635>

Chen, D. and Porada, J.H. and Hopper, J.B and Clark, N.A. “Chiral heliconical ground state nanoscale pitch in a nematic liquid crystal of achiral molecular dimers” Proceedings of the National Academy of Sciences, 110(40), 15931-15936 (2013) DOI: <https://doi.org/10.1073/pnas.1314654110>

4. The schemes of the $-1/2$ defects in Fig.4 are wrong: The director field does not diverge at the three tips; there are no letters "X" at the tips, please check any textbook on liquid crystals for a correct drawing.

The schemes for all the defects have been updated. Now a blue round marker indicates the position of the defect, while the dashed white lines are used to give a general sense of the nematic director. The following line has been added to the text: "Blue dots indicate the position of the defect and dashed white lines give an indication of the direction of the nematic director field." To the figure description. The updated figure is now:

5. Fig 5 caption: "aa" apparently means "an"

This has been corrected.

Reviewer #2:

First off, this manuscript read really well, and the topic is at the forefront of current research in condensed matter. The system interrogated by the authors, namely bent silica rods (BSR) that allow for dye labeling as well as the creation of responsive core-shell BSR structures, shows some intriguing properties that permit imaging at the single particle as well as the assembly level. Images confirm the formation of the elusive N_{SB} phase by EM and FC imaging and tie well into theoretical considerations. The phase is expected between the N and the Sm phase and does so in the experimental system. The work is significant and should receive attention in the community. Therefore, this manuscript should be considered for publication in Nature Communication. There are, however, a few points the authors may want to consider:

We thank the reviewer for the kind words about our work and about the readability of our paper. We are especially thankful on the remarks about polydispersity, as now that is considered in our document and greatly improves the discussion on the experimental formation of N_{SB} phases.

(1) How critical are the overall dimensions of the BSR (i.e., aspect ratio and bent angle)? Do theory or experiment suggest any range for which the N_{SB} phase would show/form?

Our experiments show that N_{SB} phases are found as stable phases for silica rods with bending angles of $154^\circ \pm 8^\circ$ and $151^\circ \pm 9^\circ$ aspect ratios (L/D where $L = L_{\text{short segment}} + L_{\text{long segment}}$) 10.8 and 13.8 respectively.

Although not reported in our paper, additional experiments were performed with BSRs with $129^\circ \pm 16^\circ$ and aspect ratio $L/D \approx 8.7$ with both rod segments approximately equal in length (Figure below). However, in this system of bent rods no N_{SB} phase was observed. Contrary to the systems described in our paper, for we consistently observed I, N, N_{SB} and Sm phases. However, these results need further work to be more extensively investigated and confirmed and have not been included in the current paper. Exploring the parameter space in which BSRs form N_{SB} phases is planned in future work, and a follow up paper on how to tune the angle of BSR systems is in the making first.

In summary, from our experimental observations, N_{SB} phases appear to stabilize with BSRs with segments of different lengths with angles around 150° and aspect ratios around 10 to 14. However, the full parameter space is yet to be studied, as the study on how to fully tune the properties of our BSR building blocks is ongoing. The dimensions, angles and polydispersities for which BSRs form N_{SB} phases may be widely different than that for other materials with different properties (densities, interactions) and shapes (particle shapes not based in spherocylindrical geometries).

Figure 1. (S)TEM image of bent silica rod-like particles with dimensions of $D = 273 \pm 40$ nm (PD 14 %), $L_1 = 1.193 \pm 130$ nm (PD 10 %), $L_2 = 1205 \pm 120$ nm (PD 10 %) and bending angle $\alpha = 129 \pm 16^\circ$. Scale bars A, B) 1 μ m, and 5 μ m respectively.

(2) What is the range of polydispersity for which theory or experiment would show the N_{SB} phase? In other words, how polydisperse must and can it be?

For the experimental system of bent silica rod-like particles (BSRs), the N_{SB} phases were observed for particles with ~ 13 % polydispersity.

From computer simulations, N_{SB} phases were observed for bent spherocylinder-like particles with PD in particle length of 36%, and aspect ratio of 10 (Chiappini et al. (2019)). However, in the experiments the N_{SB} phases were observed for particles with PD of ~ 13 %, which is much lower than the PD predicted by simulations. Certainly, the polydispersity of other particle parameters (such as bending angle, particle thickness, length of each rod segment) also play an important role in experimental systems and were not included in the simulations yet. For instance, with the current BSRs system with a PD of ~ 13 %, an aspect ratio of ~ 10 and a bending angle (α) around 150° we consistently found N_{SB} phases. In contrast, no N_{SB} phases were observed for BSRs with PD of ~ 10 % mentioned earlier, equal segments, and $\alpha = 129^\circ \pm 16^\circ$. The parameter space is vast and complex.

To answer the question posed by the reviewer, comparing our results with the simulation results from Chiappini et al. (2019), implies that the polydispersity in angles and particle dimensions (thickness, different segments) might lower the minimum polydispersity needed (in particle length) predicted by simulations. This is an interesting fact and might be of special relevance when translating our results with BSRs to systems of bent core mesogens, as such systems polydispersity distribution, given their molecular nature, is widely different and can be controlled generally better than that of colloidal systems.

Considering the question posed by the reviewer, we believe that adding a statement acknowledging that the polydispersity of our system is lower than that predicted by computational methods is relevant for a reader, and the following line has been added to the first paragraph of our results section:

Interestingly, the polydispersity in length of our system (13 %) was significantly lower than that described for a N_{SB} phase by computational methods (36%) (Chiappini). However, our experimental system consists of building blocks with polydispersity also in the angle between particle segments, the length of each segment and their width. Therefore, it is likely that for experimental systems the smectic phase could be destabilized by contributions from all different polydispersities combined.

(3) Considering potential applications, it would be of tremendous value if the authors could make sine more detailed remarks about how the community can translate these finding to molecular systems.

Colloidal crystals are seen as model systems to mimic atomic and molecular crystals. The findings in the current manuscript on elusive N_{SB} phases provide an opportunity to study and understand the phases on a single particle level, which may directly translate to developments in the molecular scale as well.¹ Molecular LC are extremely challenging to image at the single building block level, contrary to colloidal (liquid) crystals. Silica colloids-based LC phases (N , Sm) have been successfully fully characterized in 3D and therefore offer a perfect platform for the characterization of N_{SB} (and other LC) phases. As we already speculated about in the paper, mixtures of bent mesogens of different lengths and/or molecular structures as inspired by the polydispersity of our colloidal systems might be able to form N_{SB} phases as well.

As the interactions between molecules are more complex and they are additionally more flexible, it is hard to extrapolate to more specific conditions, but were are not aware if mixing different length molecular bend LC molecules has already been tried. Of course, our future work might also elucidate the role of pd "bendedness" as well. With the current colloidal system, it is possible to understand the N_{SB} phases from a more simplified view, as the interactions are so much less complex. Moreover, this system also allows to measure the pitch length precisely which in turn can be related to macroscopic properties of the LC phase. Particularly with csBSRs (which can respond to the external electric and magnetic fields) systems, one may be able to tune the orientation of the individual particles in real time during experiments and characterize the effects of external fields on the N_{SB} phase.

As mentioned, some of these aspects were already mentioned in our second paragraph. Nevertheless, we have added the following line at the end of the second paragraph:

The possibility to fully characterize a N_{SB} system on a single particle level in 3D that can be manipulated by external fields opens the door to directly study the effects of size, shape, angles and their corresponding polydispersity on phase formation, switching and relaxation processes.

(4) The introductory section describes biaxial nematics but leaves the biaxial smectic-A phase formed by more blank- or brick-shaped molecules out.

The biaxial smectic-A phase is an interesting phase as it is easier to experimentally realize than the biaxial nematic phase (Teixeira, P.I.C., Osipov, M.A. and Luckhurst, G.R. Simple model for biaxial smectic-A liquid-crystal phases, Physical Review E, 73, 061708 (2006)), shows optical biaxiality and has been shown to be easily switchable by external fields (Panarin, Y.P., Tschierske, C. and Vij, J.V. Switching in a Biaxial Smectic-A-like phase, Liquid Crystals Today, 30:2, 20-25 (2021)). These applications are in line with the expected applications of biaxial nematic materials. Furthermore, Smectic-A phases have been realized not only via molecular systems but also via colloidal assembly (Yang, Y. and Chen, G. and Thanneeru, S. and He, J. and Liu, K. and Nie, Z. Synthesis and assembly of colloidal cuboids with tunable shape biaxiality, Nature Communications, 9, 4513, 2018). We understand that it is interesting for a potential reader to be able to have a literature inlet in our text to literature that relates to biaxial nematic systems as well.

Therefore, the following referenced statement has been added to the first paragraph:

Furthermore, biaxial smectic-A phases \cite{Teixeira2006}, intimately related to biaxial nematic phases and also holding potential for applications such as optical switching have been reported as well for molecular\cite{Panarin2021} and colloidal systems \cite{Yang2018}.

Aside from these suggestions to provide a little more detail on some specific aspects of this work, I see no reason why this manuscript should not be published in Nature Communications.

Great work!

Reviewer #3:

This is a well-written paper describing the experimental observation of a splay-bend nematic phase in a system of polydisperse, bent silica rods. I am no experimentalist, so cannot sensibly judge the experimental procedures, but the background looks excellent to me and the analysis of the results also looks careful and thorough, so the authors have certainly convinced me that they have found an example of this somewhat elusive phase. Furthermore the synthetic route looks very promising for future application.

As the authors point out, this is not a "first", in that a colloidal splay-bend nematic phase has already been observed (ref 33). The question is thus whether this observation is sufficiently novel to warrant publication in Nature Communications. I think that the power of the new technique is enough, in my view, to overcome this barrier and I would recommend publication after the authors have attended to the following points.

We thank the reviewer for the kind works on our work. We are specially thankful for bringing up the observation that the smectic phase looks "swirly" in our figure, as this is now mentioned in the paper. The fact that this swirly smectic phase could be linked to a so called frustrated N_{SB} phases mentioned by Chiappini et al. (2019) links nicely our results to the existing literature and gives context to potential questions raised by our figure.

1) Could the authors say how they chose the dimensions, angles, etc. of their bent rods? My guess is that this sample happened to be the one that worked, but some clarity on this would be helpful. Even knowing the properties of samples that did not give this phase would be helpful.

Indeed, the results reported in this paper describe the results for samples that did stabilize N_{SB} phases. This does not mean that only samples with our particular size and angles work, but a systematic exploration of the entire is yet to follow, as we are still working on making a broad spectrum of bending angles and particle aspect ratios (paper in progress).

(From here this answer is the same as our answer to Reviewer 2, Question 1)

Our experiments show that N_{SB} phases are found as stable phases for silica rods with bending angles of $154^\circ \pm 8^\circ$ and $151^\circ \pm 9^\circ$ aspect ratios (L/D where $L = L_{\text{short segment}} + L_{\text{long segment}}$) 10.8 and 13.8 respectively.

Although not reported in our paper, additional experiments were performed with BSRs with $129^\circ \pm 16^\circ$ and aspect ratio $L/D \approx 8.7$ with both rod segments approximately equal in length (Figure below). However, in this system of bend rods no N_{SB} phase was observed. Contrary to the systems described in our paper, for we consistently observed I, N, N_{SB} and Sm phases. However, these results need further work to be more extensively investigated and confirmed and have not been included in the current paper. Exploring the parameter space in which BSRs form N_{SB} phases is planned in future work, and a follow up paper on how to tune the angle of BSR systems is in the making first.

In summary, from our experimental observations, N_{SB} phases appear to stabilize with BSRs with segments of different lengths with angles around 150° and aspect ratios around 10 to 14. However, the

full parameter space is yet to be studied, as the study on how to fully tune the properties of our BSR building blocks is ongoing. The dimensions, angles and polydispersities for which BSRs form N_{SB} phases may be widely different than that for other materials with different properties (densities, interactions) and shapes (particle shapes not based in spherocylindrical geometries).

Figure 1. (S)TEM image of bent silica rod-like particles with dimensions of $D = 273 \pm 40$ nm (PD 14 %), $L_1 = 1.193 \pm 130$ nm (PD 10 %), $L_2 = 1205 \pm 120$ nm (PD 10 %) and bending angle $\alpha = 129 \pm 16^\circ$. Scale bars A, B) 1 μ m, and 5 μ m respectively.

2) It would be useful to know more about the properties of the neighbouring phases shown in fig. 2. For example, the smectic looks very swirly by eye! What sort of smectic is it? Indeed might it not be a smectic at all, but one of these liquid crystalline frustrated phases? More detail on this would be useful.

The reviewer correctly points out that the smectic phase does seem to present some curvature in its director field. One can speculate that maybe a frustrated N_{SB} phase that is on its way of becoming a Sm domain might still partially retain long range modulations in its director or might be affected by the modulation of the neighboring (in parameter space) a N_{SB} phase. However, as mentioned as these speculations are of yet not yet backed up by computational or theoretical support, we chose not to discuss the root of these observations. However, as the observation can be relatively easily seen in the results, we have added the following text:

Particle orientations were homogeneous in the nematic regions, whereas the smectic regions showed a degree of swirliness. It is possible that some of the smectic regions are resulting from a frustrated N_{SB} phase as mentioned in a paper on computer simulations (cite{Chiappini}), and retained part of the modulations of the N_{SB} phase, perhaps until further equilibration into an unperturbed smectic.

3) Just for the record, values of the order parameters would be helpful, certainly in the nematic and SB phases.

We agree that processing the order parameter for the different phases would be a great addition to our study. However, we believe that with our current data a measurement of the order parameter will not be precise enough to add information besides what can be qualitatively evaluated from the bare imaging results. This is because our building blocks are polydisperse and contain a kink, also polydisperse, which makes it hard to define and measure exactly the orientation of single particles. We expect that as we further develop this system, and we can fully determine particle positions and orientations in 3D, we will be able to give precise quantitative measurements of parameters such as the order parameter and the pitch of the phases. The scope of this paper was to demonstrate that N_{SB}

phases form in sediments of BSRs, which can be proven by our current resolution, but a detailed study of the phases will require resolving the single particles at sufficient quality to determine their individual configurations. This has been done previously in our group with silica spheres and silica rods without bending (Bakker et al. *Soft Matter* 12:45, 2016).

REVIEWERS' COMMENTS

Reviewer #1 (Remarks to the Author):

The manuscript is improved. The only remaining issue is that the director lines shown as dashed white lines around two $-1/2$ disclinations in Fig 4 a,b do not correspond to the actual director, especially at the periphery of the schemes, where the curved lines increase their separation from the straight lines. I attach a drawing with detailed explanation. After correction, the work could be published.

Reviewer #2 (Remarks to the Author):

From my reading, the authors addressed the comments raised in my review as well as the comments made by the other two reviewers. I am very in favor of this manuscript being published.

Reviewer #3 (Remarks to the Author):

The authors have addressed all my queries and I would now recommend publication without change.

Answer to the reviewers

Reviewer #1 (Remarks to the Author):

The manuscript is improved. The only remaining issue is that the director lines shown as dashed white lines around two $-1/2$ disclinations in Fig 4 a,b do not correspond to the actual director, especially at the periphery of the schemes, where the curved lines increase their separation from the straight lines. I attach a drawing with detailed explanation. After correction, the work could be published.

We thank the reviewer for indicating our mistake and for the attached drawing to make the point clear. Indeed, the dashed lines that give a general indication of the nematic director diverged when they should have converged. We have updated our figure accordingly:

Reviewer #2 (Remarks to the Author):

From my reading, the authors addressed the comments raised in my review as well as the comments made by the other two reviewers. I am very in favor of this manuscript being published.

We thank the reviewer for his positive contributions to our manuscript.

Reviewer #3 (Remarks to the Author):

The authors have addressed all my queries and I would now recommend publication without change.

We thank the reviewer for his positive contributions to our manuscript.